# Proto-lm: A Prototypical Network-Based Framework for Built-in Interpretability in Large Language Models

**Sean Xie**
Department of Computer Science
Dartmouth College
sean.xie.gr@dartmouth.edu

**Soroush Vosoughi***
Department of Computer Science
Dartmouth College
soroush.vosoughi@dartmouth.edu

**Saeed Hassanpour***
Department of Biomedical Data Science
Dartmouth College
saeed.hassanpour@dartmouth.edu

## Abstract

Large Language Models (LLMs) have significantly advanced the field of Natural Language Processing (NLP), but their lack of interpretability has been a major concern. Current methods for interpreting LLMs are post hoc, applied after inference time, and have limitations such as their focus on low-level features and lack of explainability at higher-level text units. In this work, we introduce `proto-lm`, a prototypical network-based white-box framework that allows LLMs to learn immediately interpretable embeddings during the fine-tuning stage while maintaining competitive performance. Our method's applicability and interpretability are demonstrated through experiments on a wide range of NLP tasks, and our results indicate a new possibility of creating interpretable models without sacrificing performance. This novel approach to interpretability in LLMs can pave the way for more interpretable models without the need to sacrifice performance. We release our code at https://github.com/yx131/proto-lm.

## 1 Introduction

In recent years, Large Language Models (LLMs) have significantly improved results on a wide range of Natural Language Processing (NLP) tasks. However, despite their state-of-the-art performance, LLMs, such as BERT (Devlin et al., 2018), RoBERTa (Liu et al., 2019) and BART (Lewis et al., 2019), are not easily interpretable. Interpretability is a crucial aspect of language models, especially LLMs, as it enables trust and adoption in various domains. To address this issue, there is a growing interest in improving model interpretability for LLMs and neural models in general.

Current interpretation methods have several limitations, such as requiring a surrogate model to be built (Ribeiro et al., 2016; Lundberg and Lee, 2017) or being applied post-hoc to each instance, separate

from the original decision-making process of the model (Shrikumar et al., 2016, 2017; Springenberg et al., 2014). These limitations add extra computational complexity to the explanation process and can result in approximate explanations that are unfaithful[1] to the original model's decision-making process (Sun et al., 2020; DeYoung et al., 2019). Finally, current interpretability methods focus on attributing importance to different words in the input and do not explain the model's decision at the sentence or sample level. (Sundararajan et al., 2017; Vaswani et al., 2017; Murdoch et al., 2018).

To address these challenges, we propose a novel framework that utilizes a prototypical network to learn an interpretable prototypical embedding layer on top of a fine-tuned LLM, which can be trained end-to-end for a downstream task. Our framework utilizes trainable parameters, called prototypes, to both perform the downstream task by capturing important features from the training dataset and provide explanations for the model's decisions via projection onto the most influential training examples. Additionally, we utilize a token-level attention layer before the prototypical layer to select relevant parts within each input text, allowing our model to attribute importance to individual words like existing interpretability methods. Our framework, named `proto-lm`, achieves competitive results on a wide range of NLP tasks and offers interpretability while addressing the previously mentioned limitations of current interpretability methods.

Fig. 1 shows our proposed `proto-lm` framework applied to a multi-classification example from the SST5 dataset (Socher et al., 2013). As the figure illustrates, the model's decision-making process is simple, transparent, and accurate as it classifies the input as "Very Positive" based on its highest

---

*Co-corresponding Authors.

[1]Faithfulness is defined as how accurately the explanation reflects the decision-making process of the model (Jacovi and Goldberg, 2020).

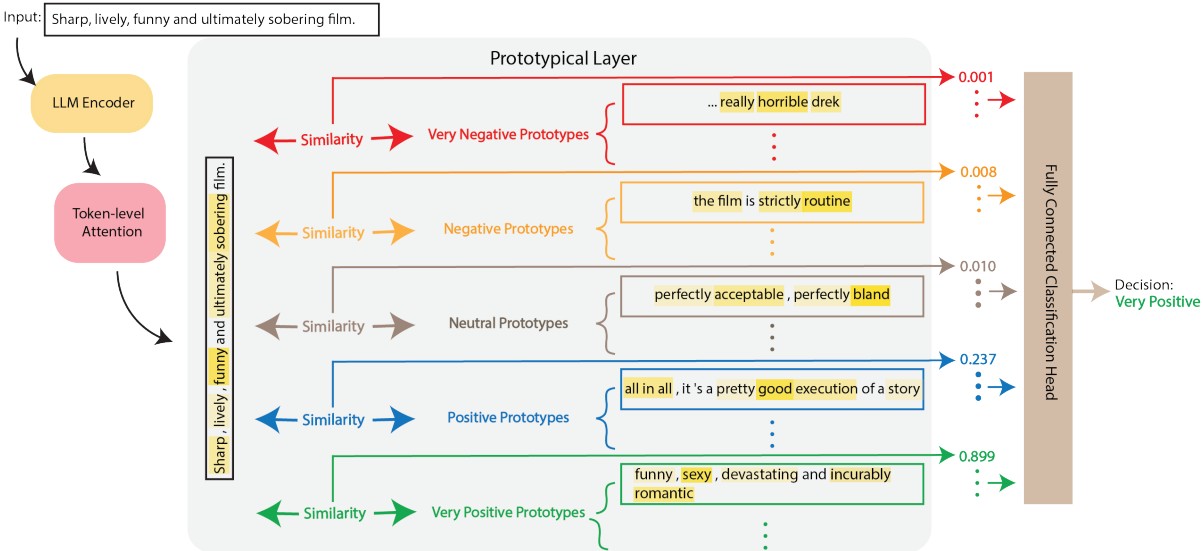

Figure 1: Illustration of the inherently interpretable decision-making process of the `proto-lm` model for an example from the SST5 (5-class fine-grained sentiment analysis) dataset. The figure highlights the words identified by token-level attention, with a stronger shade indicating a higher attention score. `proto-lm` correctly predicts the input by relating it to prototypes of the "Very Positive" class. The decision process of `proto-lm` is transparent, and no separate interpretability method is required.

similarity to prototypes from the "Very Positive" class and the input's distance (lack of similarity) to "Negative" and "Very Negative" prototypes. A key advantage of our framework is its *inherent interpretability*. Our prototypical network does not require an additional surrogate model for explanation, and we can observe the exact contribution of each prototype to the final output, resulting in faithful explanations. Additionally, our framework offers interpretability at both the token and sample level, allowing for the identification of important words in the input text as well as the attribution of importance to impactful samples in the training data. In this work, we make the following contributions:

- We introduce a novel framework, `proto-lm` based on prototypical networks that provides inherent interpretability to LLMs.
- We demonstrate `proto-lm`'s applicability on three LLMs and show its competitive performance on a wide range of NLP tasks.
- We conduct ablation studies to demonstrate important characteristics of `proto-lm` under different hyperparameters.
- We evaluate the interpretability of `proto-lm` under several desiderata and show that the explanations provided by `proto-lm` are of high quality.

## 2 `proto-lm`

The architecture of our proposed framework, `proto-lm`, is shown in Figure 2. We utilize a pre-trained LLM as the underlying language model to encode the input and build a prototypical layer on top of it to provide interpretable prototypes. The goal of the learned prototypes is to capture features representative enough of each class so that they can be fine-tuned for the downstream classification task. In the classification process, similarities between the encoded input and the learned prototypes are fed through a fully-connected layer to produce logits. We formalize this below.

### 2.1 Token-level Attention

Let $x_i$ represent a single input text and $y_i$ its associated label. We denote $f(x_i)$ as the encoding of $x_i$ by an underlying LLM, $f$. We pass $f(x_i)$ through a token-level attention layer that learns to emphasize different parts of the text, allowing us to identify important words *within* each sample. We first calculate $v_t$, the output of a fully-connected layer with bias ($\psi$), applied to $h_t$, which the embedding of each token $t$ in $f(x_i)$, followed by a $\tanh$ activation function ($\sigma$). We then compute $\nu_t$, the dot product of $v_t$, and a token-level weight vector $W_\nu$.

$$v_t = \sigma(\psi(h_t)) \qquad \nu_t = v_t \cdot W_\nu \qquad (1)$$

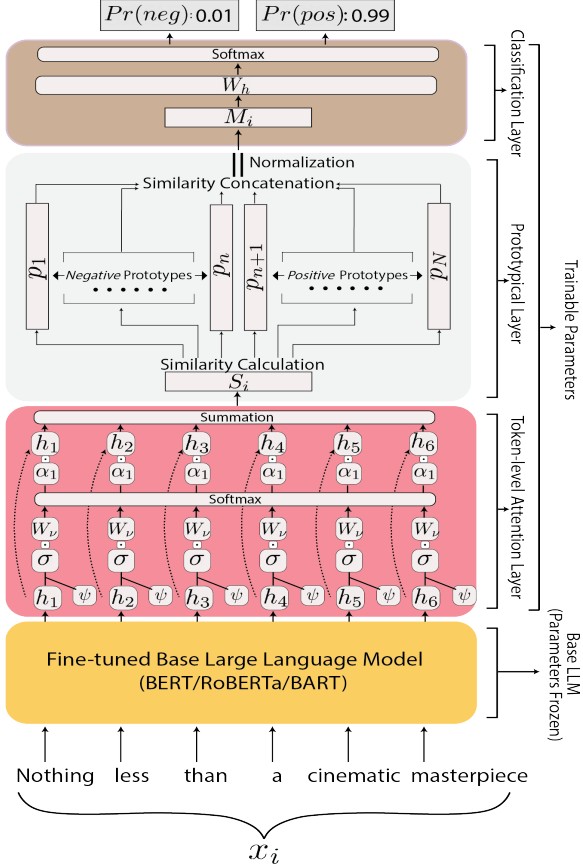

Figure 2: The protoypical network-based architecture of `proto-lm`.

We calculate the attention score $\alpha_t$ for each token $t$ using the softmax function. The attended encoding of $f(x_i)$, $S_i$, is then obtained by taking the weighted sum of the token embeddings, where the weight for each token is determined by its attention score.

$$\alpha_t = \frac{\exp(\nu_t)}{\sum\limits_{t \in f(x_i)} \exp(\nu_t)} \quad S_i = \sum\limits_{t \in f(x_i)} \alpha_t h_t \quad (2)$$

### 2.2 Prototypical Layer

In the prototypical layer $P$, we create $N$ prototype vectors and assign $n$ prototypes to each class $c \in C$, where $C$ represents the set of all classes in the dataset. To ensure an equal number of prototype vectors are allocated to capture representative features for each class, we set $n = \frac{N}{|C|}$, where $|C|$ denotes the total number of classes in the dataset. The input $S_i$ is then fed into $P$, where we calculate the similarity between $S_i$ and every prototype vector $p \in P$ by inverting the $L_2$-distance. We then concatenate all $N$ similarities to obtain the vector $M_i$, which serves as the embedding of input

$x_i$ in the prototypical space. Each dimension of $M_i$ can be interpreted as the similarity between $M_i$ and a prototype. Subsequently, $M_i$ is fed through a fully-connected layer $W_h$ of dimension $N \times |C|$ to obtain logits for each class. Let $W_c$ denote the set of weights in $W_h$ that connect similarities in $M_i$ to the logit of class $c$. Our model's output probabilities are computed as follows:

$$Pr(\hat{y} = c \mid x_i) = \frac{\exp(M_i w_c)}{\sum\limits_{c \in C} \exp(M_i w_c)} \quad (3)$$

### 2.3 Training Objective

To create more representative prototypes and shape the clusters of prototypes in the prototypical embedding space, we introduce a cohesion loss term $\mathcal{L}_{coh}$ and a separation loss term $\mathcal{L}_{sep}$ into our loss function in addition to the cross entropy loss $\mathcal{L}_{ce}$. Let $K$ be an integer hyperparameter, $p_j$ denote a singular prototype, and $P_{y_i}$ represent all prototypes in $P$ that belong to class $y_i$, our loss terms are defined as follows:

$$\mathcal{L}_{coh} = \frac{1}{K} \cdot \sum\limits_{\forall j: p_j \in P_{y_i}} \max_K \|S_i - p_j\|_2^2 \quad (4)$$

$$\mathcal{L}_{sep} = -\frac{1}{K} \cdot \sum\limits_{\forall j: p_j \notin P_{y_i}} \min_K \|S_i - p_j\|_2^2 \quad (5)$$

For every $S_i$ in the input batch, $\mathcal{L}_{coh}$ penalizes the average distance between $S_i$ and the $K$ most distant prototypes of its class ($y_i$) while $\mathcal{L}_{sep}$ penalizes the average distance between $S_i$ and the $K$ most similar prototypes that do not belong to $y_i$. Intuitively, for every $S_i$, $\mathcal{L}_{coh}$ "pulls" $K$ prototypes of the *correct* class close while $\mathcal{L}_{sep}$ "pushes" $K$ prototypes of the *incorrect* class away. We then add cross-entropy loss $\mathcal{L}_{ce}$ and weight each loss term with a hyperparameter $\lambda$ such that $\lambda_0 + \lambda_1 + \lambda_2 = 1$ to obtain the following loss function:

$$\mathcal{L}_{total} = \lambda_0 \cdot \mathcal{L}_{ce} + \lambda_1 \cdot \mathcal{L}_{coh} + \lambda_2 \cdot \mathcal{L}_{sep} \quad (6)$$

### 2.4 Prototype Projection

To understand each prototype vector $p_j$ in natural language, we project each prototype onto the nearest token-level attended encoding ($S_j$) of a sample text ($x_j$) in the training data that belongs to the same class as $p_j$ and assign the token-attended text of that prototype. Let $D$ be the training

dataset. We formally denote the projection as in eq.7: $\forall (x_j, y_j) \in D$ such that $y_j = c$:

$$\text{Text of } p_j \leftarrow \underset{S_j}{\operatorname{argmin}} \|S_j - p_j\|_2^2 \qquad (7)$$

In other words, for each prototype $p_j$, we find the nearest $S_j$ in the training dataset that belongs to the same class as $p_j$ and assign the corresponding token-attended text to $p_j$ (Details in App. D & E).

## 3   Performance Experiments

As we do not want interpretability to come at the cost of performance, we first conduct experiments to assess the modeling capability of `proto-lm`. We implement `proto-lm` using BERT (base-uncased and large-uncased), RoBERTa (base and large) and BART-large as the base LLM encoders and train the token-level attention module, prototypical layer and classification head as described in §2. We evaluate the predictive accuracy of `proto-lm` on 5 tasks (SST-2, QNLI, MNLI, WNLI, RTE) from the GLUE dataset (Wang et al., 2018), IMDb sentiment analysis (Maas et al., 2011) and SST-5 (Socher et al., 2013). For baselines, we use the same fine-tuned versions of the LLMs with a classification head. We tune all hyperparameters using the respective validation data in each dataset. We present the mean performance as well as the standard deviation over 5 runs under their respective optimal configurations of hyperparameters (cf. App.A) and compare them to their baselines in Table 1. Across our experiments [2], we observe that `proto-lm` either closely matches or exceeds the performance of its baseline LLM, proving that there is not a trade-off between `proto-lm`'s interpretability and performance.

## 4   Prototype Interpretability

### 4.1   Interpretable prototypical space and decision-making

Compared to black-box models such as BERT/RoBERTa/BART, which have uninterpretable embedding dimensions, `proto-lm` offers the added benefit of interpretability in conjunction with competitive performance. We show an example of a 2-dimensional prototypical space in Fig. 3, where `proto-lm` provides insight into the model's decision-making process by allowing

---

[2]We conduct additional performance experiments with `proto-lm` in App.B

---

us to visualize examples in prototypical space, where each dimension represents the normalized similarity $\in [0, 1]$ between an example and one prototype.

In Fig. 3, we select one positive and one negative prototype from the prototypical layer of the model to create a 2-dimensional space in which we place examples #1-#4. The vertical axis represents similarity to the negative prototype, and the horizontal axis represents similarity to the positive prototype. From this, we can see that example #1 is much more similar to the negative prototype than to the positive prototype, and example #3 is much more similar to the positive prototype than to the negative prototype, and both are correctly classified as a result.

From the prototypical space, we can see clearly that the model's decision to misclassify example #2 as positive is due to elements in the example that make it similar to the positive prototype. Similarly, the prototypical space of `proto-lm` reveals that example #4 was misclassified because, while it contains elements that are similar to both the positive and negative prototypes, it is closer to the negative prototype. The interpretable prototypical space of `proto-lm` provides an explanation for why difficult cases such as examples #2 and #4 are misclassified. It should be noted that while similar analyses and explanations can be obtained through post-hoc techniques such as generating sentence embeddings (Reimers and Gurevych, 2019; Gao et al., 2021), `proto-lm` has this capability built-in.

### 4.2   Prototype quality and performance

We investigate the effect of different weightings of the loss terms in our loss function. We train `proto-lm`, holding all other hyperparameters constant, under 11 different values of $\lambda_0$ evenly distributed on $[0, 1]$, and report our results in Figure 4. For these experiments, we place equal weight on $\mathcal{L}_{coh}$ and $\mathcal{L}_{sep}$ such that $\lambda_1 = \lambda_2 = \frac{1-\lambda_0}{2}$ (additional details in App. C).

Because prototypes not only help interpret the model but also capture the most representative features from the data, placing an excessive emphasis on $\mathcal{L}_{ce}$ actually achieves the adverse effect. As shown in Figure 4, increasing $\lambda_0$ comes at the cost of learning representative prototypes (which $\mathcal{L}_{coh}$ and $\mathcal{L}_{sep}$ do) and this is reflected in decreasing classification accuracy on the downstream task. We

| | SST2 | QNLI | MNLI | WNLI | RTE | IMDb | SST5 |
|---|---|---|---|---|---|---|---|
| proto-lm/BERT-base | **93.6** ±0.02/93.2 | 90.5 ±0.02 /90.2 | 84.0 ±0.03 /84.6 | **47.9**/46.4* | **68.1** ± 0.01/66.4 | **91.5** ± 0.02/91.3 | **55.3** ± 0.03/54.9 |
| proto-lm/BERT-large | **95.2** ±0.03/94.9 | 92.4 ±0.03 /92.7 | 86.3 ±0.02 /86.7 | 47.9/47.9* | **76.2** ± 0.02/70.1 | **93.6** ± 0.03/92.0 | **56.4** ± 0.02/56.2 |
| proto-lm/RoBERTa-base | **94.0** ±0.03/93.6 | 92.2 ±0.01 /92.5 | 86.9 ±0.03 /87.2 | 56.3/56.3* | **84.2** ± 0.03/78.7 | **95.0** ± 0.02/94.7 | **56.8** ± 0.03/56.4 |
| proto-lm/RoBERTa-large | **96.5** ±0.02/96.4 | 94.6 ±0.02 /94.7 | 90.1 ±0.02 /90.2 | 56.4/56.4* | **88.6** ± 0.02/86.6 | **95.8** ± 0.03/95.6 | **58.0** ± 0.04/57.9 |
| proto-lm/BART-large | **97.0** ±0.04/96.6 | 94.2 ±0.05 /94.9 | 89.3 ±0.06 /90.2 | - | **89.2** ± 0.05/87.0 | - | - |

Table 1: Predictive accuracy of `proto-lm` compared against fine-tuned versions of their base LLM's. We observe competitive performance when compared to baselines on all tasks, with bolded and underlined values representing better and equivalent performances, respectively. * indicates that the performance is a result of our experiments on that task and not reported in the original work cited. All results for `proto-lm` are results under the optimal arrangement of hyperparameters for that task, which we obtain through experimentation.

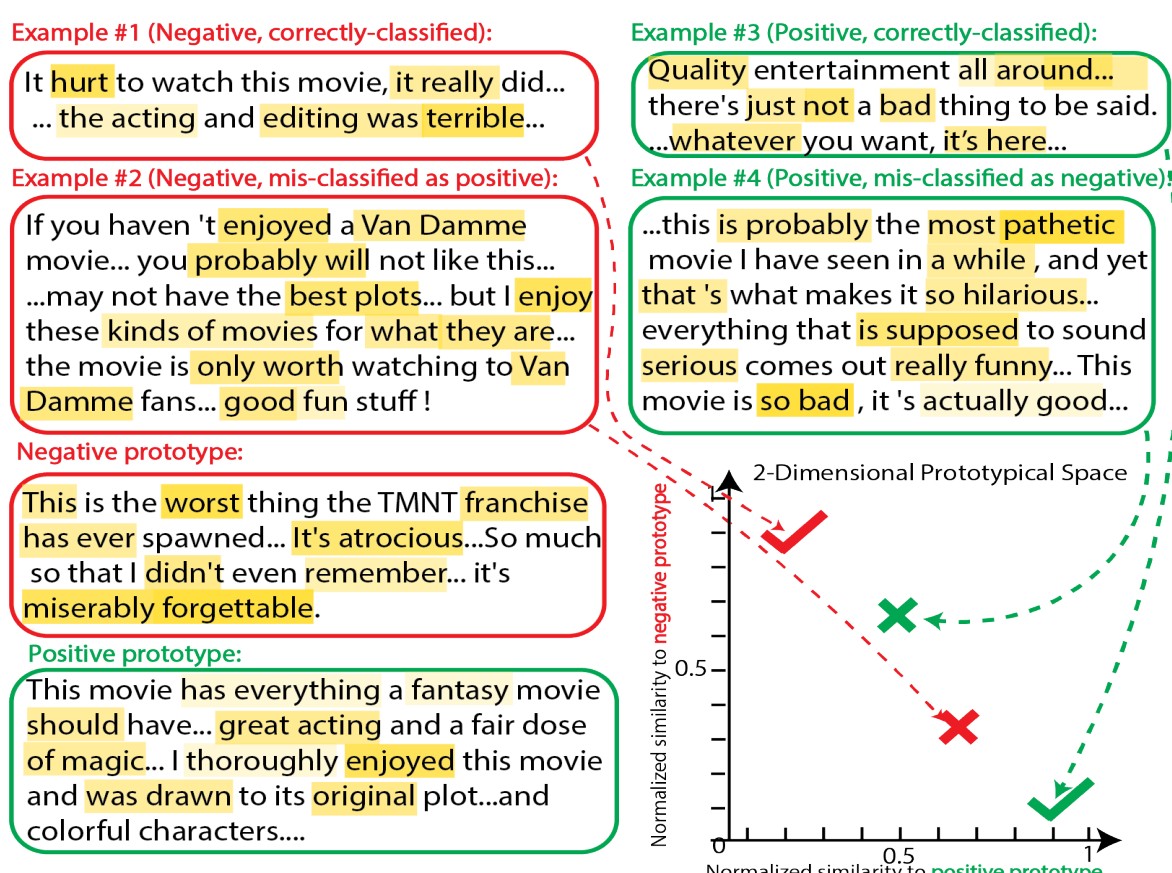

Figure 3: Two-dimensional prototypical space representation of four test examples, where each dimension represents the normalized similarity between the sample and a prototype. A positive prototype is chosen for the horizontal axis, and a negative prototype for the vertical axis. Examples #1 and #3 are correctly classified, as they are much more similar to the prototype of their respective ground-truth class. Examples #2 and #4 are misclassified, due to their high similarities to the prototype of the incorrect class.

observe that the optimal accuracy is achieved when $\lambda_0 = 0.3$. As we increase $\lambda_0$ from 0.3, we see not only a decline in accuracy but also a decline in the quality of the prototypes associated with the input. On the other, placing sole emphasis on $\mathcal{L}_{coh}$ and $\mathcal{L}_{sep}$ without any emphasis on $\mathcal{L}_{ce}$ (as in the case of $\lambda_0 = 0$) leads to prototypes that do not help with the downstream task.

### 4.3 Size of prototypical space

As noted in §4.2, prototypes serve to capture features from data that are useful for making predictions in addition to interpretability. As a result, the number of prototypes ($N$) in the prototypical layer directly affects the expressiveness of our model. We conduct experiments varying the number of prototypes $N$ using RoBERTa-Large as the base and show our results in Fig. 5. We observe a general increase in accuracy up until $N = 1000$, after which

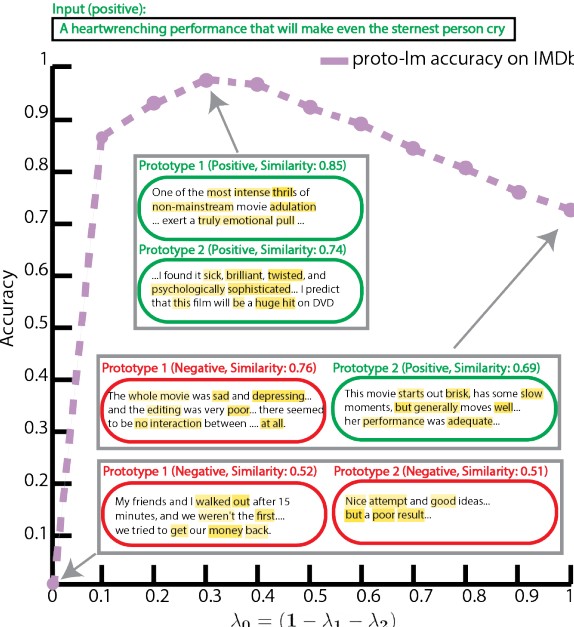

Figure 4: Predictive accuracy of the `proto-lm` model with RoBERTa-large as the base LM on the IMDb dataset as a function of $\lambda_0$, with $N = 1400$, $n = 700$, and $K = 350$. We illustrate the closest (most similar) prototypes to the positive input generated by `proto-lm` when trained under different values of $\lambda_0$. We observe a non-linear relationship between accuracy and $\lambda_0$.

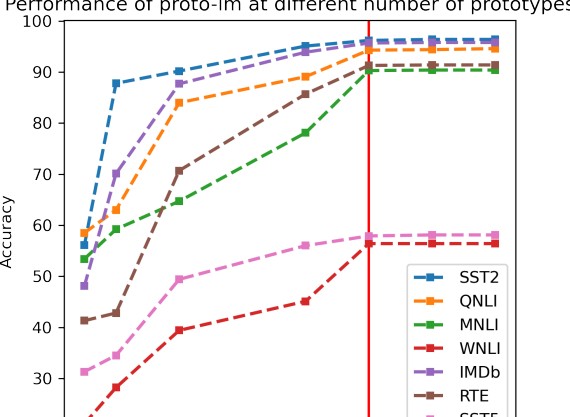

Figure 5: Performance of `proto-lm` on tasks under varying numbers of prototypes in the prototypical layer ($N$).

accuracy plateaus for some tasks while increasing only slightly for others. We reason that increasing $N$ can only improve the model's expressiveness until the point when $N$ reaches the embedding dimension of the underlying LLM, after which more prototypes in the prototypical space no longer aid in capturing more useful features from the training data. In Fig. 5's case, as RoBERTa-large's output dimension is 1024, we see the increasing trend of accuracy stop at around 1000 prototypes.

### 4.4 Prototype uniqueness and performance

The interpretability of `proto-lm` stems from the retrieval of the most informative training examples. If all prototypes are predominantly associated with a limited number of training examples, this reduces their utility from an interpretability perspective. Hence, it is beneficial to encourage prototype *segregation*, that is, a broader projection onto the dataset and a more diverse representation of different samples. Besides normalizing prototype distances, which has been shown to influence prototype segregation (Das et al., 2022), `proto-lm` introduces an additional hyperparameter, $K$. This parameter controls the number of prototypes that each training example associates

and disassociates with during training. As changing $K$ also influences the decision-making process of the model by altering the number of samples the models compare for each input, we examine the impact of $K$ on both prototype segregation and model performance. In our experiment, we employ `proto-lm` with RoBERTa-large as our base on SST2 and IMDb. We set $N$, the total number of prototypes, to be 1000, and $n = 1000/2$, the number of prototypes per class, to be 500. We train models under seven different $K$ values, keeping all other hyperparameters constant. We report the accuracy of the models and the percentage of **unique** prototypes in $P$ under each different $K$ in Fig.6.

A prototype is considered unique if no other prototype in $P$ projects onto the same sample in the dataset. We observe that the best prototype segregation (highest number of unique prototypes) occurs when $K = 1$, and the number of unique prototypes significantly drops as we increase $K$. Intuitively, if more prototypes are drawn close to each sample during training (eq.4), it becomes challenging to create unique prototypes. It is important to note that eq. 5 is less related to the uniqueness of prototypes as prototypes are projected onto samples from the *same* class. We also witness a slight rise in model accuracy as we increase $K$. We conjecture that while unique prototypes contribute to interpretability, the model doesn't necessarily require prototypes to be unique to make accurate decisions. Thus, we observe a minor trade-off between interpretability and model performance.

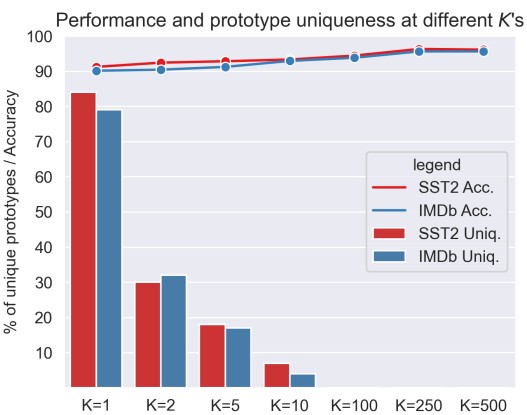

Figure 6: Performance and prototype uniqueness in `proto-lm` when trained with different $K$'s.

# 5 Evaluating the Interpretability of `proto-lm`

Through the usage of its prototypical space, `proto-lm` is a white-box, inherently interpretable model. But just how well do the explanations provided by `proto-lm` satisfy common desiderata in interpretability? We conduct experiments to try to answer that question in this section. Specifically, we evaluate the inherent interpretability provided by `proto-lm` via measures of faithfulness (DeYoung et al., 2019; Jacovi and Goldberg, 2020) and simulatability (Pruthi et al., 2022; Fernandes et al., 2022).

## 5.1 Faithfulness experiments

Recently, the concept of faithfulness, which measures the extent to which an explanation accurately reflects the true decision-making process of a model, has garnered attention as a criterion for evaluating explainability methods (Jacovi and Goldberg, 2020, 2021; Chan et al., 2022; Dasgupta et al., 2022). For textual inputs, faithfulness is concretely evaluated using the metrics of *comprehensiveness* (Comp) and *sufficiency* (Suff), as defined by DeYoung et al. (2019). Specifically, Comp and Suff quantify the reduction in the model's confidence for its output when salient features are *removed* and *retained*, respectively.

We extend the application of Comp and Suff to prototypical networks to assess the faithfulness of `proto-lm`. For an input $x_i$, we initially identify the top $k\%$ of most similar prototypes and the bottom $k\%$ of least similar prototypes. Let $p_i^k$ denote the $k\%$ prototypes identified, we compute Comp as the

percentage difference in model confidence when $p_i^k$ are *removed* (by setting $p_i^k$'s weights in $W_h$ are set to 0). Specifically, let $\hat{y}_i$ be the prediction of a model $\mathcal{M}$ on $x_i$, let $pr_{\hat{y}_i}$ be the output logit of $M$ for $\hat{y}_i$, and let $pr_{\hat{y}_i}(P \setminus p_i^k)$ be the output logit when $p_i^k$ prototypes are removed from the prototypical layer, we calculate Comp as follows:

$$\text{Comp} = \frac{pr_{\hat{y}_i} - pr_{\hat{y}_i}(P \setminus p_i^k)}{pr_{\hat{y}_i}} \qquad (8)$$

We analogously calculate Suff as the percentage difference in model confidence when the $k\%$ of prototypes are *retained*:

$$\text{Suff} = \frac{pr_{\hat{y}_i} - pr_{\hat{y}_i}(p_i^k)}{pr_{\hat{y}_i}} \qquad (9)$$

We compute Comp and Suff $k \in 1, 5, 10, 20, 50$ and our present mean results across SST2, QNLI, MNLI and SST5 in Fig 7. Our formulation of prototype Comp and Suff are inspired by DeYoung et al. (2019). We note here that under this formulation, a **lower** sufficiency is *better* i.e., a smaller amount of reduction in model confidence when only salient features are retained is better. As we remove more top $k\%$ prototypes, we observe a general increase in Comp. We also note a general decrease in Suff (a lower Suff is preferable) as we retain more top $k\%$ prototypes. Moreover, when the bottom $k\%$ of prototypes are removed, there are relatively small changes in Comp and large changes in Suff when only the bottom $k\%$ of prototypes are retained. These trends underscore the influence of the learned prototypes on the model's decision-making process and their faithfulness.

## 5.2 Simulatability experiments

Simulatability refers to the capacity of a human to mimic or replicate the decision-making process of a machine learning model (Doshi-Velez and Kim, 2017; Pruthi et al., 2022). It is a desirable attribute as it inherently aligns with the goal of transparently communicating the model's underlying behavior to human users (Fernandes et al., 2022). We assess the simulatability of `proto-lm` by providing human annotators with various explanations of model outputs on SST2/QNLI and recording the percentage of model outputs that the annotators can replicate. We provide explanations under the following four settings:

- **No Explanations (NE)**: Annotators are presented with only the sample data, without any explanations, and asked to make a decision.

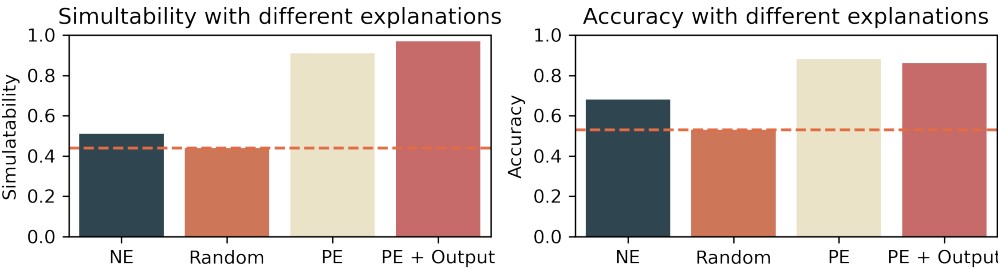

Figure 7: The comprehensiveness and sufficiency of `proto-lm` with $k\%$ prototypes removed.

Figure 8: Simulatability and accuracy of human annotators under different settings of provided explanations from `proto-lm`.

- **Random Explanations (Random)**: Each sample in the dataset is presented along with prototypes from `proto-lm` chosen *randomly* as explanations. This setting serves as our baseline.
- **Prototypical Explanations (PE)**: Each sample in the dataset is presented along with the top 5 prototypes most similar to the sample when `proto-lm` made its decision.
- **PE + Output**: In addition to the prototypical explanations, the model decision for each sample is also presented.

We employ workers from Amazon Mechanical Turk to crowdsource our evaluations and presented 3 workers with 50 examples each from the SST2 and QNLI datasets, along with explanations for the model's decisions in the settings mentioned in §5.2. We use the best performing models for SST2 and QNLI (those presented in Table 1), since previous studies found that the utility of PE's are reliant on model accuracy (Das et al., 2022). Additionally, we assess the accuracy of the human annotators in relation to the ground truth labels. We present results for both in Fig. 8. The results show that the case-based reasoning explanations offered by PEs are more effective in assisting annotators in simulating model predictions than the random baseline. We also notice a minor drop in accuracy when we provide PE + output compared to just PE. We

attribute this to the fact that the models are not entirely accurate themselves, and in instances where the model is inaccurate, presenting the output leads annotators to replicate inaccurate decisions.

Furthermore, we compare `proto-lm`'s simulatability against 3 other well-known interpretability methods: LIME, Integrated Gradients, and Guided Backprop. We employed three workers for each example and reported the mean percentage of examples where the workers correctly replicated the model's decision. For an example of the a simulatability questionnaire with PE, see Fig.12. For an example of an accuracy questionnaire with Random explanations, see Fig.13. For an example of the questionnaire for LIME/IG/GB explanations, see Fig.14. For results of `proto-lm` against LIME, IG and GB, please see Table 2. Our addditional results further indicate that `proto-lm` allow the workers to replicate the model's decisions better than all the other interpretability methods on both tasks.

## 6 Related Works

Prototypical networks (Snell et al., 2017) have shown strong performance in few-shot image and text classification tasks (Sun et al., 2019; Gao et al., 2019). However, these approaches do not actively learn prototypes and instead rely on summarizing

|                          | SST-2  | QNLI   |
|--------------------------|--------|--------|
| Random Explanations      | 42.3%  | 48.7%  |
| LIME                     | 87.3%  | 90.3%  |
| Integrated Gradient      | 84.7%  | 84.3%  |
| Guided Backprop          | 78.0%  | 88.0%  |
| **Prototype explanations from `proto-lm`** | **90.0%** | **92.0%** |

Table 2: Simulatability of `proto-lm` on SST-2 and QNLI reported as the mean percentage of model decisions replicated by three workers.

salient parts of the training data. Their focus is on learning *one* representative prototype per class while their performance is dependent on the size of the support set in few-shot learning scenarios. Due to these limitations, there have been relatively few works that utilize prototypical networks to provide interpretability for LLM's in NLP (Garcia-Olano et al., 2022; Das et al., 2022; Van Aken et al., 2022). Chen et al. (2019) and Hase et al. (2019) use prototypical parts networks with multiple learned prototypes per class but only apply their methods to image classification tasks.

Our work is most closely related to Das et al. (2022) and Van Aken et al. (2022) in terms of the approach taken. However, our work differs in that the architecture in (Das et al., 2022) only utilizes a single negative prototype for binary classification, while `proto-lm` enables multi-class classification by using multiple prototypes for each class. Additionally, we have extended the work of (Das et al., 2022) by implementing token-level attention to identify not only influential samples but also influential sections of text within each sample. Moreover, different from the single-prototype-as-a-summary approach in Van Aken et al. (2022), by learning multiple prototypes per class, `proto-lm` creates a prototypical space (§4.1), where, unlike the embedding space of LLM's, each dimension is meaningful, specifically the distance to a learned prototype, and can be used to explain a decision. Our work is also similar to Friedrich et al. (2021) in terms of our loss function design. However, Friedrich et al. (2021) 's loss function aims to maximize the similarity of the closest prototypes of the same class. Conversely, our approach strives to minimize the distance of the furthest prototypes of the same class. This results in Friedrich et al. (2021)'s approach tending to draw a single prototype closer to a specific example, potentially lim-

iting prototype diversity and representation power. Friedrich et al. (2021) counteracts this potential limitation by introducing an additional diversity loss term. `proto-lm`, in contrast, ensures prototype diversity by leveraging the K hyperparameter, which we delve into in section §4.4.

# 7 Conclusion

We introduce `proto-lm`, a white-box framework designed to offer inherent interpretability for Language Model Learning (LLMs) through interpretable prototypes. These prototypes not only explain model decisions but also serve as feature extractors for downstream tasks. Our experimental results indicate that `proto-lm` delivers competitive performance across a range of Natural Language Processing (NLP) tasks and exhibits robustness under various hyperparameter settings. The interpretability of `proto-lm` is evaluated, and our findings show that `proto-lm` delivers faithful explanations that can effectively assist humans in understanding and predicting model decisions.

# 8 Limitations

While `proto-lm` offers inherent interpretability by creating a connection between input text and pertinent parts of training data through the use of prototypes, it remains dependent on an underlying language model to convert text into a semantic space. Consequently, the interpretability of `proto-lm` is constrained by the interpretability of the foundational language model.

Moreover, interpreting the significance of a learned prototype in the context of Natural Language Processing (NLP) tasks is still an open research area. Computer Vision (CV) methods used for visualizing prototypes in image-based tasks, such as upsampling a prototypical tensor, are not transferrable to language embeddings. Instead, researchers depend on projecting prototypes onto nearby training examples to decode prototype tensors into comprehensible natural language.

# 9 Ethics Statement

We present `proto-lm`, a framework that enhances the interpretability of Language Model Learning (LLMs) by providing explanations for their decisions using examples from the training dataset. We anticipate that the broad applicability of `proto-lm` across various NLP tasks will promote transparency and trust in the use of LLMs, thereby en-

couraging their wider adoption. As observed by authors like Rudin (2019) and Jiménez-Luna et al. (2020), models with higher levels of interpretability inspire more confidence and enjoy greater utilization. We aim to contribute to advancing the field of interpretability research in NLP through our work.

For assessing the simulatability of our method, we employed Amazon Mechanical Turk (MTurk) for our human evaluation. To ensure English proficiency among workers, we restricted their location to the United States. Furthermore, only workers with a HIT approval rate of at least 99% were permitted to undertake our tasks. We provided a compensation of $0.20 per task, which roughly translates to about $24 per hour, significantly exceeding the US federal minimum wage. To uphold anonymity, we refrained from collecting any personal information from the annotators.

## 10 Acknowledgements

This research was supported in part by grants from the US National Library of Medicine (R01LM012837 & R01LM013833) and the US National Cancer Institute (R01CA249758). In addition, we would like to extend our gratitude to Joseph DiPalma, Yiren Jian, Naofumi Tomita, Weicheng Ma, Alex DeJournett, Wayner Barrios Quiroga. Peiying Hua, Weiyi Wu, Ting Shao, Guang Yang, and Chris Cortese for their valuable feedback on the manuscript and support during the research process.

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

## A Training Details

In our experiments, we utilized two NVIDIA Titan RTX and two GeForce RTX 3090 GPUs to run our experiments. We conducted an extensive search for the best hyperparameters through experimentation. Our models were trained for a maximum of 40 epochs. We initialize weights (in the classification layer) connecting each prototype's similarity to the logit of their respective weights to 1 and other weights to -.5. This technique (Chen et al., 2019) allows for better association of prototypes and faster convergence. In terms of initializing the parameters within the prototypes, we initialized them randomly from [0,1]. We used a learning rate and batch size to be 3e-6 and 128, respectively. We employed Adam (Kingma and Ba, 2014) as our optimizer with $\beta$'s of $(0.9, 0.999)$. Additionally, we limited the input size (number of tokens) to 512 during tokenization. We also investigated the optimal number of $N$ and tested proto-nlp under $N \in \{100, 200, 400, 800, 1000, 1200, 1400\}$. We discuss thsi in §4.3. Similarly, we tested $K \in \{1, 2, 5, 10, 100, 250, 500\}$ and reported the results in §4.4. Furthermore, we evaluated the effect of $\lambda_0, \lambda_1$, and $\lambda_2$ on our framework. The results of one of these tasks (IMDb) are presented in Figure 4 in the paper. The optimal $\lambda_0$ for the remaining tasks are reported in Table.3.

| | $\lambda_0$ | $\lambda_1$ | $\lambda_2$ |
|---|---|---|---|
| SST2 | 0.2 | 0.4 | 0.4 |
| QNLI | 0.1 | 0.45 | 0.45 |
| MNLI | 0.1 | 0.45 | 0.45 |
| WNLI | 0.2 | 0.4 | 0.4 |
| RTE | 0.3 | 0.35 | 0.35 |
| IMDb | 0.3 | 0.35 | 0.35 |
| SST5 | 0.2 | 0.4 | 0.4 |

Table 3: Optimal $\lambda$'s for each task found through experiments.

We utilized pretrained transformer models from Hugging Face, including:

- BERT-base-uncased:
  https://huggingface.co/bert-base-uncased

- BERT-large-uncased:
  https://huggingface.co/bert-large-uncased

- RoBERTa-base:
  https://huggingface.co/roberta-base

| | HoC (F1) | BIOSSES (MSE) |
|---|---|---|
| proto-lm/PubMedBERT | 83.15 ± 0.88/82.32 | 1.32 ± 0.14 / 1.14 |
| proto-lm/BioGPT | 82.78 ± 0.43/85.12 | 1.16 ± 0.10 / 1.08 |

Table 4: proto-lm's performance on biomedical datasets HoC and BIOSSES

- RoBERTa-large:
  https://huggingface.co/roberta-large

- BART-large: https://facebook/bart-large

## B Additional experiments and interpretability examples

We perform additional experiments with proto-lm on two biomedical datasets: The Hallmarks of Cancer Corpus (HoC) (Baker et al., 2016) and Sentence Similarity Estimation System for the Biomedical Domain (BIOSSES) (Soğancıoğlu et al., 2017). HoC is a text classification dataset comprising of 1852 abstracts from PubMed publications that have been annotated by medical experts based on a taxonomy. BIOSSES consists of 100 pairs of PubMed sentences, with each pair having been evaluated by five expert annotators. The sentences are assigned a similarity score ranging from 0 (indicating no relation) to 4 (indicating equivalent meanings).

We build proto-lm with two pretrained LLM's for the biomedical domain: PubMedBERT (Gu et al., 2021) and BioGPT (Gu et al., 2021). We report our results on the biomedical datasets in Table 4. For the HoC results, we use $N = 1000$, $\lambda_0 = 0.3$, $\lambda_1 = 0.35$, $\lambda_2 = 0.35$, and $K = 1$. For the BIOSSES results, we use $N = 25$, $\lambda_0 = 0.6$, $\lambda_1 = 0.25$, $\lambda_2 = 0.25$, and $K = 1$. Also, as BIOSSES is a regression task, we set $C$ to 1, forgo the softmax layer in the classification head, and replace $\mathcal{L}_{ce}$ with $\mathcal{L}_{mse}$. Similar to our results in Table. 1, we observe competitive performances from proto-lm, once again showing that proto-lm offers interpretability, but not at the cost of performance. To demonstrate proto-lm's interpretability, we additionally show 3 examples from HoC and the most/least similar prototypes found for those examples in Fig. 11.

## C Effect of unequal $\lambda_1$ and $\lambda_2$

We conduct experiments on proto-lm, varying only $\lambda_1$ and $\lambda_2$ (the weights of coh and sep losses, respectively) while keep other hyperparameters the same. We show the results for these instances of proto-lm on SST5 in Figure. 9. We observe that

while differing $\lambda_1$ and $\lambda_2$ values lead to convergence, placing equal emphasis on $\lambda_1$ and $\lambda_2$ leads to convergence at a lower loss. Intuitively, relying on either only pulling the correct prototypes together (cohesion) or only relying on pushing the incorrect prototypes apart (separation) is sufficient to create a meaningful prototypical space that allows for adequate performance on the task. Our experimental results show that placing equal emphasis leads to better performance.

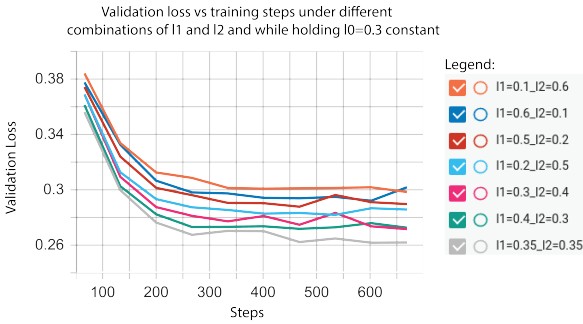

Figure 9: Loss of `proto-lm` on SST5 with different combinations of $\lambda_1$ and $\lambda_2$. We observe faster convergence at a lower loss when $\lambda_1 == \lambda_2$.

## D   Projecting prototypes

In order to help human users understand prototypes in natural language, we identify the closest training data sample for each prototype and project the prototype onto that sample. The projection of prototypes onto the nearest sample is a well-studied and established technique (Das et al., 2022; Van Aken et al., 2022; Chen et al., 2019; Hase et al., 2019). We compare the quality of our projections against the projections obtained via the training procedure of Proto-tex (Das et al., 2022) and the loss function used in Protoypical Network (Chen et al., 2019) by measuring the average normalized distance between each prototype and their projected sample's embedding. We show the results for two datasets in Fig. 10. We observe that `proto-lm` is able to train prototypes that are much closer in distance to their projected samples' embeddings than the prototypes obtained via Protoypical Network loss and within a margin of error to that of Proto-tex.

## E   Prototype faithfulness

In addition to experiments in §5.1 and §5.2, we provide a theoretical justification for the inherent interpretability in `proto-lm` in this section. Denote $d_i^j$ as the distance between two prototypes and/or

Figure 10: Average normalized distance between prototypes and the output embedding of the prototypes' projected samples.

training samples $i$ and $j$. Let $\Pi_j = \pi_{1:|D|}^j$ be the probability distribution for prototype $p_j$ over the dataset $D$, where $\pi_i^j = \frac{\eta_j}{d_i^j}$ and $\eta_j$ is a constant for normalization. Let $a$ and $b$ represent two training samples, their relative probabilities (for being projected onto by $p_j$) would be $\frac{\pi_a^j}{\pi_b^j} = \frac{\eta_j/d_a^j}{\eta_j/d_b^j} = \frac{d_b^j}{d_a^j}$. In addition, $\sum_{k \in D} \pi_k^j = 1 = \sum_{k \in D} \eta_j/d_k^j \rightarrow \eta_j = \frac{1}{\sum_{k \in D} 1/d_k^j}$, which means that each prototype is a soft-clustering over examples in $D$. Moreover, since $d_a^j/d_b^j = \frac{1}{\pi_a^j/\pi_b^j}$, if a $a$ is $n$ times further away from $p_j$ than $b$, then $b$ is $n$ times more probable in the probability distribution $\Pi_j$. Similarly, during inference time, for example $k$, if a prototype $i$ is $n$ times away from $k$ than prototype $j$, then $d_k^i/d_k^j = n \rightarrow \pi_k^j/\pi_k^i = n$, i.e. $j$ will be $n$ times more probable to be the prediction than $i$ for $k$.

## F   Interpretability examples and sample Mturk questionnaires

| Input Example | Predicted class | Most similar prototypes | <- class | Least similar prototypes | <-class |
|---|---|---|---|---|---|
| In the absence of T(regs) , the avidity of the primary immune response was impaired, which resulted in reduced memory to Listeria monocytogenes | 10 Avoiding immune destruction (ID) Immune system' Cancer Immunosuppression Cancer | Although no direct NDMA-related cancer was reported in humans , our data point to a potential epigenetic carcinogenicity of nitrosamines due to chronic immunosuppression . | 10 Avoiding immune destruction (ID) Immune system' Cancer Immunosuppression Cancer | Overall , the results provide evidence for enhanced toxicological responses in fish following exposure to Cu either alone or in combination with hypoxic condition and lends support to the evolving viewpoint that many water quality guidelines should be revisited.. | 7 Genome instability & mutation (GI) Mutation Cancer 'DNA repair' Cancer |
| | | The data also indicate that sequence homology and/or chemical properties to the original epitope are not the sole determining factors for the observed immunostimulatory activity of the mimic peptides | 10 Avoiding immune destruction (ID) Immune system' Cancer Immunosuppression Cancer | Fiberoptic bronchoscopy did not reveal any important pathologic findings | 7 Genome instability & mutation (GI) Mutation Cancer 'DNA repair' Cancer |
| | | CCL2 and LCN2 cooperatively generate immunoregulatory dendritic cells ( DCreg ) having suppressive activity accompanied by lowered expression of costimulatory molecules such as HLA-DR but increased expression of immunosuppressive molecules such as PD-L1 in human PBMCs | 10 Avoiding immune destruction (ID) Immune system' Cancer Immunosuppression Cancer | Here we show that AMPK negatively regulates aerobic glycolysis ( the Warburg effect ) in cancer cells and suppresses tumor growth invivo | 3 Resisting cell death (CD) Apoptosis Cancer Necrosis Cancer Autophagy Cancer |
| PDGFR-α and β were found to be over-expressed in the ependymoma tumor cells in seven out of 24 cases ( 29.2 % ) | 8 Tumor-promoting inflammation (TPI) Inflammation Cancer 'Oxidative stress' Cancer Inflammation 'Immune response' Cancer | Angiogenic activity present in the conditioned media of inflamed human rheumatoid synovial tissue macrophages or lipopolysaccharide-stimulated blood monocytes was equally blocked by antibodies to either IL-8 or tumor necrosis factor-alpha | 8 Tumor-promoting inflammation (TPI) Inflammation Cancer 'Oxidative stress' Cancer Inflammation 'Immune response' Cancer | RESULTS Plasma CAT activity was increased in B-CLL patients compared with control subjects ; also , progression of disease was related with significantly higher plasma activity of CAT | 1 Sustaining proliferative signaling (PS) Proliferation Receptor Cancer 'Growth factor' Cancer 'Cell cycle' Cancer |
| | | In the hyperplastic mucosa adjacent to KM12SM tumor in the cecum of athymic mice , VEGF upregulation was associated with hypoxia-inducible factor ( HIF)-1 alpha induction | 8 Tumor-promoting inflammation (TPI) Inflammation Cancer 'Oxidative stress' Cancer Inflammation 'Immune response' Cancer | The mechanism of accelerated cellular senescence was not activated by either compound in PC3 or lymph node carcinoma of the prostate ( LNCaP ) cells | 2 Evading growth suppressors (GS) 'Cell cycle' Cancer 'Contact inhibition' |
| | | Systemic treatment with C225 not only reduced tumor growth and the number of blood capillaries but also hindered the growth of established vessels toward the tumor | 8 Tumor-promoting inflammation (TPI) Inflammation Cancer 'Oxidative stress' Cancer Inflammation 'Immune response' Cancer | Furthermore , the combination of lovastatin with gefitinib induced a potent apoptotic response without significant induction of autophagy that is often induced during metabolic stress inhibiting cell death` | 4 Enabling replicative immortality (RI) Senescence Cancer Immortalization Cancer |
| Tumor necrosis was present in 10 cases ; eight of these patients died of disease and one is alive with disseminated metastases . | 5 Inducing angiogenesis (A) Angiogenesis Cancer 'Angiogenic factor' | Ectopic expression of miR-34a decreased tumor cell invasion and metastasis , inhibited the formation of promigratory cytoskeletal structures , suppressed activation of the RHO GTPase family and regulated a gene expression signature enriched in cytoskeletal functions and predictive of outcome in human lung adenocarcinomas | 5 Inducing angiogenesis (A) Angiogenesis Cancer 'Angiogenic factor' | To analyze the contribution of glycolysis to the energy supply during apoptosis , experiments were carried out with cells deprived of glucose | 3 Resisting cell death (CD) Apoptosis Cancer Necrosis Cancer Autophagy Cancer |
| | | It inhibited some metastatic properties of A431 cells supressing colony formation by soft agar assay and inhibition of MMP 9 activity by gelatin zymography and western blot analysis | 5 Inducing angiogenesis (A) Angiogenesis Cancer 'Angiogenic factor' | MiRNAs also have been detected in the blood of cancer patients and can serve as circulating biomarkers | 7 Genome instability & mutation (GI) Mutation Cancer 'DNA repair' Cancer |
| | | Increased Snail expression induces EMT and the CSC-like phenotype in CRC cells , which enhance cancer cell invasion and chemoresistance | 5 Inducing angiogenesis (A) Angiogenesis Cancer 'Angiogenic factor' | In addition , overexpression of miR-29 resulted in more rapid cell cycle re-entry from quiescence . | 9 Deregulating cellular energetics (CE) Glycolysis Cancer 'Warburg effect' Cancer |

Figure 11: Example inputs from HoC, their predictions and prototypes identified by `proto-lm`

Sentiment Analysis:
A model has made a decision on the sentiment (positive/negative) for the following movie review (#30).
This model made its decision on this review's sentiment by matching this review with 5 other reviews that
it considers very similar to this review.
**What was the model's decision?** (positive/**negative**)

## Review #30:

[BEGIN]  could easily be called the best korean film of 2002  [END]

- - - - - - - - - - - - - - - - - - - - - - - - - - - - - - - - - - - - - - - - - - - - - - - - - - - - - -

## 5 matched reviews:

Matched Review #1: [BEGIN] impresses as a  skillfully a-
ssembled ,  highly polished and professional adaptation [END]

The sentiment of this review was: **positive**

Matched Review #2: [BEGIN] the film 's intimate camera work
[END]

The sentiment of this review was: **positive**

Matched Review #3: [BEGIN] of masterpiece [END]

The sentiment of this review was: **positive**

Matched Review #4: [BEGIN] this intricately structured and
well-realized drama [END]

The sentiment of this review was: **positive**

Matched Review #5: [BEGIN] utterly absorbing [END]

The sentiment of this review was: **positive**

Figure 12: Example Amazon Mechanic Turk questionnaire used to evaluate simulatability with prototypical explanations (PE)

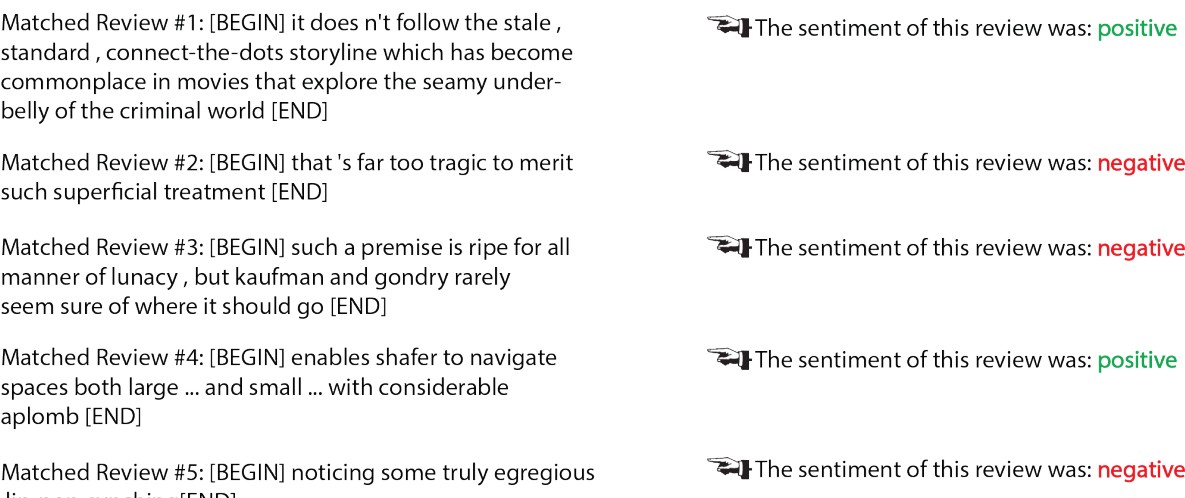

Sentiment Analysis:
**What is the sentiment of the following review (#48)?** (positive/**negative**)
You may use the matched reviews below to help you make your decision

## Review #48:

[BEGIN]  proves once again he has n't lost his touch , bringing off a superb performance in an admittedly middling film  [END]

- - - - - - - - - - - - - - - - - - - - - - - - - - - - - - - - - - - - - - - - - - - - - - - - - - -

A model has made a decision on the sentiment of review #48 by matching it with reviews it considers to be similar to #48. Here are 5 of the matched reviews:

Matched Review #1: [BEGIN] it does n't follow the stale , standard , connect-the-dots storyline which has become commonplace in movies that explore the seamy under-belly of the criminal world [END]

The sentiment of this review was: positive

Matched Review #2: [BEGIN] that 's far too tragic to merit such superficial treatment [END]

The sentiment of this review was: negative

Matched Review #3: [BEGIN] such a premise is ripe for all manner of lunacy , but kaufman and gondry rarely seem sure of where it should go [END]

The sentiment of this review was: negative

Matched Review #4: [BEGIN] enables shafer to navigate spaces both large ... and small ... with considerable aplomb [END]

The sentiment of this review was: positive

Matched Review #5: [BEGIN] noticing some truly egregious lip-non-synching[END]

The sentiment of this review was: negative

Figure 13: Example Amazon Mechanic Turk questionnaire we used to evaluate accuracy with random explanations.

Example MTurk questionnaire with explanations provided by LIME/Integrated Gradients/ Guided Backprop

Choose the primary sentiment that is expressed by the following movie review.
Please pay close attention to the highlighted words to help you make your decision.
Words highlighted in green carry positive sentiments. Words highlighted in red carry negative sentiments.
The darker the shade the stronger the sentiment.
You may not agree with the colors and sentiments that certain words carry. You will have a chance to reflect your thoughts on this in the answers.

Legend: ■ Negative  □ Neutral  ■ Positive

## Review #25:

[BEGIN] is so deadly dull that watching the proverbial paint dry would be a welcome improvement . [END]

Figure 14: Example Amazon Mechanic Turk questionnaires we used to evaluate simulatability of LIME/Integrated Gradients/Guided Backprop.