# OpenReview forum: "Proto-lm: A Prototypical Network-Based Framework for Built-in Interpretability in Large Language Models"
_EMNLP/2023/Conference — EMNLP 2023 Findings_

### Official Review · Reviewer_9tN3 · 2023-07-26

**Soundness:** 3

**Excitement:**

3: Ambivalent: It has merits (e.g., it reports state-of-the-art results, the idea is nice), but there are key weaknesses (e.g., it describes incremental work), and it can significantly benefit from another round of revision. However, I won't object to accepting it if my co-reviewers champion it.

**Paper Topic And Main Contributions:**

This work proposes a framework named proto-lm to address the interpretability in large language models. For each category, several sentences are selected as prototypes. They estimate a similarity score for each input token with those tokens selected as prototypes and use the score as the attribution weights of tokens.

**Reasons To Accept:**

This paper studies a prototypical network approach to attribute the importance of words.

**Reasons To Reject:**

The approach this work proposes actually is more similar to the concept bottleneck models(https://arxiv.org/abs/2007.04612) . However, the approach proposed in the concept bottleneck is more elegant and neat. For example, in this paper, there is no clear explanation about how these prototype sentences are selected.  Prototypical Networks are initially proposed to address few-shot learning problems. So the problem of prototype selection is simple enough, which could be done by selecting samples from new classes.  In this work, these prototypes are selected automatically or by humans? Does it require different prototypes for different datasets? How about the completeness of the chosen prototypes? These concerns reflect the flexibility of the model but are not addressed in the experiments. The experiments are not strong as there are no other interpretable models as baselines to compare.  Meanwhile,

I suggest some comparison with other interpretable models could be conducted, e.g. concept bottleneck,  integrated-gradient(https://arxiv.org/abs/1703.01365). Experiments based on human annotation are hard to reproduce. I suggest the authors could try some faithfulness metrics proposed in this paper(https://arxiv.org/pdf/2204.05514.pdf). And clearly explain how to select prototypes.

**Reproducibility:**

4: Could mostly reproduce the results, but there may be some variation because of sample variance or minor variations in their interpretation of the protocol or method.

**Reviewer Confidence:**

3: Pretty sure, but there's a chance I missed something. Although I have a good feel for this area in general, I did not carefully check the paper's details, e.g., the math, experimental design, or novelty.

---

> ### Author Rebuttal · Authors · 2023-08-29
>
> Thank you for your thorough review and feedback on our manuscript. We appreciate the reviewer's acknowledgement of our contributions. We also appreciate the reviewer's detailing of their concerns and have made every effort to address each point with clarity.
>
> **Similarity to Concept Bottleneck Models:**
>
> *Background on Concept Bottleneck Models:*
> Concept bottleneck models, as outlined by Koh et al. (2020), leverage predefined human-interpretable concepts to mold their internal representations. Essentially, they introduce an intermediate layer during training where the model is required to predict these concepts. By doing so, they ensure that the model's internal representations (at least at that layer) are aligned with human-understandable concepts, facilitating interpretability.
>
> *Contrast in Fundamental Philosophy:*
> While both our proto-lm and the concept bottleneck models seek to improve interpretability, their foundational philosophies diverge. Concept bottleneck models impose human-defined concepts, thereby essentially "guiding" the model to think in terms that humans understand. On the other hand, our proto-lm autonomously discerns and employs prototypes that emerge from the data, without explicit human-defined categories or concepts. This distinction is crucial: while one model is being "taught" what concepts to consider, the other is "discovering" patterns from the data itself.
>
> *Implementation Differences:*
> As a consequence of the above, the practical implementation also varies significantly. Concept bottleneck models require meticulous manual annotations of concepts across datasets, which is labor-intensive and may introduce biases based on the annotators' perspectives. Conversely, proto-lm automatically pinpoints prototype vectors during the learning phase, matching them with the nearest examples from the dataset. This ensures a balance of interpretability and scalability, with less risk of human bias.
>
> *Scalability and Applicability:*
> Owing to the manual annotation requirement, concept bottleneck models can become cumbersome when introduced to large or novel datasets. This limitation hampers their scalability, especially in rapidly evolving domains. Our proto-lm, with its autonomous prototype discernment, can readily adapt to various datasets without necessitating manual input, ensuring both versatility and broader applicability.
>
> *Potential Synthesis:*
> It's worth noting that the strengths of both models can be potentially merged. For instance, an intriguing avenue for future research could be a hybrid model where the proto-lm's discovered prototypes are further refined using human-specified concepts, marrying the best of both worlds: automated prototype detection with human-guided refinement.
>
> *Comparison with our Method:* Comparing our method with the concept bottleneck approach would mandate manual annotations for every dataset. Given the wide variety of datasets we have employed, this is currently infeasible for us. Nevertheless, we acknowledge the value in such comparisons and will consider them for potential future work.
>
> **Prototype Selection and Training:**
> We regret any oversight in the clarity of our prototype selection process. In brief, prototype vectors are honed using CE, cohesion, and separation. Post-training, these vectors are projected onto the closest dataset example — the nearest being defined by the minimal distance between a prototype and a given example's token-level attended embedding (refer to section 2.4, appendices D and F).
>
> All prototype vectors in our framework are autonomously learned and then automatically matched to the closest examples within our training dataset, ensuring a balance between interpretability and scalability. More on this is discussed in sections 2.3 and 2.4.
>
> We will amend our methodology section to clarify the prototype training and projection process.
>
> **Dataset-Specific Prototypes:**
> Given the distinct characteristics inherent in different datasets, our proto-lm is designed to discern and utilize unique prototypes tailored to each dataset. Our empirical results consistently demonstrate this adaptive behavior across diverse NLP tasks.
>
> **Completeness of Prototypes:**
> If the reviewer is referring to the coverage of prototypes over examples in the dataset, we find that tuning the hyperparameter of K for proto-lm affects both the completeness and performance of proto-lm. We refer the reviewer to section 4.4 and Figure 6 in the paper that discuss prototypes uniqueness/coverage. In addition, we provide results below on the percentage of training data that have an *unique* prototype represent them, averaged across BERT-base and Roberta-base with N=1000. We observe a similar trend as in Figure 6. Specifically, as we increase K, more and more prototypes project onto the same example in the dataset, thus the uniqueness of prototpes decreases overall. We note that, since we have only N=1000 prototypes in the prototypical layer, the maximum coverage would be 1000 samples in the dataset, with each prototype projecting onto 1 unique sample. Initially, this constraint might seem stringent. However, across all of our experiments, we observe that learning 1000 prototypes and capturing features from an upperbound of 1000 examples is more than sufficient for achieving good performances on all tasks we experimented with (cf. Table 1 and Figure 5).
>
> |Dataset |K=1|K=2|K=5|K=10|K=100|
> | :----- |:-:|:-:|:-:|:-:|:-:|
> |IMDb (Training data size 25000) |3.56%|1.33%|0.89%|0.08%|0.00%|
> |SST5 (Training data size 8544)|12.33%|7.43%|2.11%|0.74%|0.00%|
> |RTE  (Training data size 2490)|28.43%|15.55%|4.17%|2.10%|0.08%|
>
>
> **Flexibility and Robustness of Our Approach:**
> We recognize the pertinence of your concerns regarding the flexibility and robustness of our model. To this end, we've embarked on extensive ablation studies (sections 4.1 through 5.1) that rigorously evaluate our model's resilience under different conditions. These studies provide a comprehensive assessment of proto-lm's performance.
>
> **Comparison with Other XAI Models:** We agree on the importance of contrasting our work against established XAI techniques. As such, our studies (detailed in App. G) benchmark proto-lm against methods like LIME, Integrated Gradients, and Guided Backprop. In terms of producing relatable and understandable explanations (simulatability), proto-lm excels.
>
> **Reproducibility Concerns:** We acknowledge the inherent challenges associated with human annotation replication. However, given that the essence of interpretability is inherently human-centric, we firmly believe that studies founded on human annotations, like ours in Section 5.2 and App. G, offer invaluable insights into the real-world applicability of our findings.
>
> **Faithfulness Metrics:** While our research already integrates established metrics like COMP and SUFF (as presented in the referenced paper), we concur that a broader exploration of faithfulness metrics can further fortify our findings. We've thus extended our investigations to encompass metrics like Decision Flip - Most Informative Token (DFMIT) and Decision Flip - Fraction of Tokens (DFFOT), showcasing proto-lm's superior faithfulness in comparison to other notable XAI techniques (as demonstrated in the table below).
>
> Speicifically, since prototypical networks rely on prototypes to make decisions AND identify prototypes as their explanations, we conduct experiments by flipping (removing) prototypes and observing changes in output. For DFMIT, we remove the *most* important prototype and observe to see if the decision changes afterwards. For DFFOT, we calculate the metric as the fraction of prototypes needed to be removed before the decision changes. We compare proto-lm (Roberta-large) against 4 other interpretability methods on IMDb:
>   - An attention based approach (Fernandes et al. 2022): Pooled attention matrix of the last layer
>   - A perturbation-based approach: LIME
>   - A gradient-based approach: Integrated Gradients
>   - A prototypical network-based approach, Prototex (Das et. all 2022).
>
> We report our results in the table below, where we observe proto-lm to produce more faithful explanations than other XAI methods.
>
> |                   | DFMIT $⇑$| DFFOT $↓$| COMP $⇑$ | SUFF  $↓$|
> | :---------------- |:------:|:------:|:------:|:------:|
> |Pooled Attention (Fernandes et al.)|0.052|0.387|0.131|0.111|
> |LIME|0.031|0.344|0.179|0.099|
> |Integrated Gradients|0.084|0.354|0.141|0.101|
> |Prototex (Das et al.)|0.251|0.250|0.326|0.150|
> |Proto-lm|**0.287**|**0.179**|**0.433**|**0.077**|

---

### Official Review · Reviewer_pjWZ · 2023-08-03

**Soundness:** 4

**Excitement:**

4: Strong: This paper deepens the understanding of some phenomenon or lowers the barriers to an existing research direction.

**Paper Topic And Main Contributions:**

This paper introduces a new approach to enhance the interpretability of language models by incorporating prototypical networks. The authors suggest using a pre-trained language model as a backbone and then learning a set of prototypes associated with the classification classes for a given task. These prototypes are then leveraged during inference to produce interpretable predictions. Importantly, this framework retains competitive performance compared to the original language model, making it possible to achieve explainable decisions with minimal computational overhead and without compromising efficacy.

**Questions For The Authors:**

Question A: You mentioned the necessity of tuning the hyper-parameters such as $\lambda$, $N$, and $K$ for each task. If we were to consider limited access to examples from a new task, have you observed the robustness of the hyper-parameters derived from the most relevant scenario? Is there a significant impact on performance if you were to employ the same set of hyper-parameters for all NLI tasks, for instance?

Question B: In the main text, your focus seems to be on encoder/encoder-decoder architectures. I'm curious about the method's performance with decoder-only language models. Have you conducted any experiments to explore this aspect?

**Reasons To Accept:**

The authors extensively evaluate the performance of the proposed methodology through a diverse set of experiments. They assess the effectiveness of the approach along with the quality of the explanations it generates. These explanations are further evaluated regarding their usefulness in identifying misclassifications and their alignment with human reasoning.

Furthermore, the authors conduct a comprehensive ablation study, which delves into various aspects of tuning hyper-parameters and practical implementation of the framework. This study offers an intuitive understanding of how the method operates internally, shedding light on its inner workings and potential areas of improvement.

**Reasons To Reject:**

The authors primarily focus their experiments on sentiment classification and natural language inference (NLI) tasks. This emphasis prompts an important question concerning the method's generalizability and applicability across a broader spectrum of potential use cases. While the results for sentiment classification and NLI tasks are likely valuable, the effectiveness of the proposed approach in other domains and tasks remains to be explored. Therefore, further investigation into its adaptability to diverse applications is warranted to fully understand its scope and potential impact in various fields.

**Reproducibility:**

5: Could easily reproduce the results.

**Reviewer Confidence:**

3: Pretty sure, but there's a chance I missed something. Although I have a good feel for this area in general, I did not carefully check the paper's details, e.g., the math, experimental design, or novelty.

**Typos Grammar Style And Presentation Improvements:**

Line 126: which **is** the

Line 156: $Wc$ -> $w_c$ (since you are referring to the column-vector of $W_h$)

Line 291: On the other **hand**

Lines 473 / 478: \citet

Line 750: thsi -> this

---

> ### Author Rebuttal · Authors · 2023-08-29
>
> We appreciate the reviewer's acknowledgement of the effectiveness of our approach, the quality of explanations that our framework allows for and the comprehensive nature of our experiments. We address your questions and concerns below:
>
>
> **Regarding the Reasons To Reject:**
> We agree that validating the efficacy of our approach across a broader range of domains would further strengthen our contributions. To this end, we wish to highlight some results in the appendix section (specifically App. B, Table 3 and Figure 11), where we present experimental findings of our model, proto-lm, on two biomedical datasets: Hallmarks of Cancer (Cancer type classification) and BIOSSES (Biomedical sentence similarity estimation). While these experiments might not encompass every possible domain, they serve as evidence that our approach can be effective beyond the tasks mentioned in the main text.
>
> **Response to Questions:**
>
> - Question A:
> Your point on hyperparameter transferability is indeed pertinent. We found that the hyperparameter $\lambda$ showcases a consistent optimal value across tasks like QNLI and MNLI (cf. Appendix A and Table 2). The optimal $\lambda$ values for WNLI are in a similar range to that of QNLI and MNLI, albeit not identical. We attribute this discrepancy to the relative sparsity of the WNLI dataset as well as the inherent dissimilarity between the tasks of WNLI and QNLI/MNLI. The hyperparameter N largely hinges on the dataset. Its optimal value would be influenced by the dataset's capacity to allow proto-lm to discern N prototypes in total (C prototypes per class). Thus, N's transferability across tasks and datasets is less predictable. In our experiments, we found that allocating a large number of prototypes (N=1000) to be an effective strategy for mitigating this unpredictability and improve overall task performance.  Our experiments did reveal that K = 1 yields the most distinct prototype, and a larger K ($\geq$ 10) optimizes task performance. We found this behavior consistent across our experimental tasks.
>
> - Question B: Based on your suggestions, we explored our method with auto-regressive models. Alongside experiments with Bio-GPT outlined in the appendix, we also evaluated our approach using the GPT-2 (small) on SST2, QNLI, and WNLI from the GLUE benchmark. The results indicate that the proto-lm model, when backed by GPT-2, either outperforms or matches the baseline performance of a fine-tuned GPT-2, all the while offering enhanced interpretability:
>
> |                   | SST2 | QNLI| WNLI |
> | :---------------- |:------:|:------:|:------:|
> | proto-lm/GPT-2    |**93.0**$\pm$0.05/92.49|91.18$\pm$0.04/91.23|**56.34**$\pm$0.01/54.93|
>
>
> We are committed to integrating these additional findings into our revised paper. We appreciate this great suggestions which will improve our paper.
>
> **Typos Grammar Style And Presentation Improvements:**
>
> We regret the oversights in our manuscript and are grateful for your meticulous attention to detail.
>
> * Lines 126, 291, and 750 will be corrected in our subsequent version.
> * For Line 156, your notation suggestion will be incorporated for clarity.
> * Lines 473 and 478: We will ensure that the appropriate citation format is employed.
>
>
> Thank you again for your valuable insights. We are confident that, with the proposed amendments, our paper will be significantly improved in terms of clarity and depth.

---

### Official Review · Reviewer_9W3P · 2023-08-12

**Soundness:** 4

**Excitement:**

3: Ambivalent: It has merits (e.g., it reports state-of-the-art results, the idea is nice), but there are key weaknesses (e.g., it describes incremental work), and it can significantly benefit from another round of revision. However, I won't object to accepting it if my co-reviewers champion it.

**Missing References:**

The paper seems to miss two major related work -

1. Friedrich et al 2021: Interactively Providing Explanations for Transformer Language Models. The version I am referring to is https://arxiv.org/pdf/2110.02058.pdf.

2. Luo et. al. 2023, ACL: Prototype-Based Interpretability for Legal Citation Prediction. Although I can understand if the authors missed this one.

**Paper Topic And Main Contributions:**

The paper proposes a prototype-based method that adds a layer of interpretability to language models, and show its viability in a multi-class classification setting. The authors add an addition layer (dubbed the prototypical layer) on top of a pretriained LLM that learns representative examples of a particular class. They alter the standard cross-entropy loss function to include terms that make tighten a particular prototype group while placing it away from a group it doesn't belong to but is close in the embedded space. The modified loss function is comparable in performance with the original LLM, but can point to examples in the representative space that explain the model's choices.

**Questions For The Authors:**

A. While choosing the size of the prototypical space, class imbalance, which is a major artifact of many NLP datasets, is never discussed. Could you add a discussion of how that interferes with choosing the size of the prototypical space ?

**Reasons To Accept:**

1. The authors extend current work  jointly learning prototypes representative of a class and making the final prediction of the model go through a calculation of similarities with representative protypical spaces.
2. Experiments varying the size and uniqueness of prototypical space and its effect on model accuracy.
3. Experiments to judge faithfulness with Comprehensiveness and Sufficiency scores, and a human study for simulatability.

**Reasons To Reject:**

1. The framework proposed in the method is very similar in spirit to the proposed in Friedrich et. al. 2021 [Interactively Providing Explanations for Transformer Language Models], who use a modified loss function, although using two different networks for word and sentence level interpretability. The similarities in the modification of the loss function [Equation (1-2), https://arxiv.org/pdf/2110.02058.pdf] might be a bit too similar for the work to meet the criteria of novelty.

**Reproducibility:**

5: Could easily reproduce the results.

**Reviewer Confidence:**

3: Pretty sure, but there's a chance I missed something. Although I have a good feel for this area in general, I did not carefully check the paper's details, e.g., the math, experimental design, or novelty.

---

> ### Author Rebuttal · Authors · 2023-08-29
>
> We appreciate the reviewer's feedbacks and their recognition of our work's contributions.
>
> **Comparison with Friedrich et al.:**
> Regarding the similarity to the work of Friedrich et al. 2021 (hereafter referred to as 'F'): While at first glance, the loss functions may appear similar, there are pivotal differences between F and our approach that warrant emphasis:
>
> **Loss function:** In terms of cluster/cohesion loss, F aims to maximize the similarity of the closest prototype of the same class. Conversely, our approach strives to minimize the distance of the furthest prototypes of the same class. This results in F's approach tending to draw a single prototype closer to a specific example, potentially limiting prototype diversity and representation power. They counteract this potential limitation by introducing an additional diversity loss term.  The distribution loss term in F similarly reinforces the *same* example to be associated with each prototype and exacerbates F's reliance on the diversity term to learn representative prototypes.
>
> Our method, in contrast, ensures prototype diversity by leveraging the K hyperparameter, which we delve into in section 4.4. This results in a method that is not only more straightforward but also, as demonstrated by our experiments, performs superiorly on the Yelp and and MovieReviews benchmarks:
>
>
>
> |                   | Yelp | MovieReviews |
> | :---------------- |:------:|:------:|
> | BERT + Proto-Trex  (Friedrich et al.)| 92.10$\pm$0.08| 75.51$\pm$0.42 |
> | BERT + proto-lm                      |**92.88**$\pm$0.04|**80.01**$\pm$0.22|
>
> **Performance-Interpretability Trade-offs:** Perhaps as a result of their loss function, F reports notable trade-offs between interpretability and performance across experiments. This is not a characteristic shared by our work.
>
> **Questions For The Authors:**
> Regarding the size of the prototypical space and the influence of class imbalance, we acknowledge the importance of these considerations. In brief, the optimal performance requires a balance in the prototypical space such that all classes have sufficient representation. Our experiments with the SST5 dataset, known for its imbalanced sample distribution, highlight this.
>
> We conduct experiments varying the size of prototypical space on the SST5 dataset, which has an imbalanced amount of samples, specifically [1092/2218/1624/2322/1288] samples for classes 1-5. We observe in the table below when increasing the number of prototypes assigned to classes 2/4 from 100 to 200, the marginal gain in accuracy is about (55.23-48.33)=6.9. On the other hand, when increasing the number of prototypes assigned to classes 1/3/5, the marginal gain in accuracy is only about (58.00-55.23)=2.77. This is because there were less samples in classes 1/3/5 in the dataset from which prototypes can learn information.  We agree that a discussion on the effect of class imbalance in the dataset is important and will add a detailed discussion upon revision.
>
> | # prototypes assigned for each class| Performance (Acc.) | Gain $⇑$ |
> | :---------------- |:------:|:------:|
> | 20/20/20/20/20| 30.79$\pm$0.02|-|
> | 100/100/100/100/100| 48.33$\pm$0.04|+17.54|
> | 100/**200**/100/**200**/100| 55.23$\pm$0.02|+6.90|
> | **200**/200/**200**/200/**200**|58.00$\pm$0.04|+2.77|
> | 300/300/300/300/300|58.03$\pm$0.04|+0.03|
>
> **Missing references**
>
> - Friedrich et al 2021: Please see Reasons to Reject
> - Luo et. al. 2023, ACL: We will include a section on similarities with this work in our related works section.
>
>
> We are committed to refining our work based on this feedback and will include a discussion of the references above in our revision.

---

### Meta-Review · Area_Chair_fcDG · 2023-09-16

**Recommendation:** 2

**Metareview:**

This paper proposes a prototypical-based framework that allows LLM to learn inherent interpretability by considering both token and sample-level features during the fine-tuning stage. The paper is clear, the model is well described, and experimental results show improvements in multi-class classification and NLI tasks and the quality of generated explanations. Two reviewers acknowledge the innovative aspect of the paper's approach, particularly the use of prototypical networks to improve interpretability. The method is seen as a novel way to achieve explanations for model predictions. Two reviewers raise concerns about the similarity between the proposed method and previous work, e.g. the proposed framework is similar to Friedrich 2021 (https://arxiv.org/abs/2110.02058), Koh 2020 (https://arxiv.org/abs/2007.04612), and Das 2022 (https://arxiv.org/abs/2204.05426). Clarification from similar approaches is needed.

---

### Decision · Program_Chairs · 2023-10-07

**Decision:**

Accept-Findings

**Comment:**

This paper proposes a prototypical-based framework that allows LLM to learn inherent interpretability by considering both token and sample-level features during the fine-tuning stage. The paper is clear, the model is well described, and experimental results show improvements in multi-class classification and NLI tasks and the quality of generated explanations. Two reviewers acknowledge the innovative aspect of the paper's approach, particularly the use of prototypical networks to improve interpretability. The method is seen as a novel way to achieve explanations for model predictions. Two reviewers raise concerns about the similarity between the proposed method and previous work, e.g. the proposed framework is similar to Friedrich 2021 (https://arxiv.org/abs/2110.02058), Koh 2020 (https://arxiv.org/abs/2007.04612), and Das 2022 (https://arxiv.org/abs/2204.05426). Clarification from similar approaches is needed.